# Digital Form for Assessing Dentistry Undergraduates Regarding Periodontal Disease Associated with Cardiovascular Diseases

**DOI:** 10.3390/medicina59030509

**Published:** 2023-03-05

**Authors:** Rebeca Antunes de Medeiros, Yngrid Monteiro da Silva, Yasmim Marçal Soares Miranda, Danyelle de Sousa Gomes, Tabata Resque Beckmann Carvalho, Erich Brito Tanaka, Paula Gabriela Faciola Pessôa de Oliveira, Jorge Sá Elias Nogueira, Silvio Augusto Fernandes de Menezes, Tatiany Oliveira de Alencar Menezes, Rogério Valois Laurentino, Ricardo Roberto de Souza Fonseca, Luiz Fernando Almeida Machado

**Affiliations:** 1School of Dentistry, University Center of State of Pará, Belém 66060-575, PA, Brazil; 2School of Dentistry, Federal University of Pará, Belém 66075-110, PA, Brazil; 3Biology of Infectious and Parasitic Agents Post-Graduate Program, Federal University of Pará, Belém 66075-110, PA, Brazil; 4Virology Laboratory, Institute of Biological Sciences, Federal University of Pará, Belém 66075-110, PA, Brazil

**Keywords:** periodontal medicine, oral and systemic disease interactions, cardiovascular disease, oral health medicine, arterial hypertension

## Abstract

*Background*: Throughout recent years, periodontal disease (PD) has been linked to innumerable medical systemic conditions, such as cardiovascular disease (CVD). This association could negatively impact oral health, so the knowledge of dentists who have graduated must follow modern dentistry in order to promote oral health, mainly in systemically compromised patients. Therefore, the present study aimed to determine and evaluate the knowledge level of dentistry undergraduate students (DUS) regarding the correct periodontal treatment and management of cardiac patients with PD. *Methods*: This cross-sectional and populational-based study was conducted between March and June 2022 in northern Brazil. A total of 153 DUS received an anonymous digital form (Google Forms Platform) using a non-probabilistic “snowball” sampling technique. The digital form was composed of four blocks of dichotomous and multiple-choice questions. After signing the informed consent term, DUS were divided into three groups according to their period/semester in dentistry graduation during the study time (G1: 1st period/semester; G2: 5th period/semester and G3: 10th period/semester). A total of 25 questions referring to demographic, educational and knowledge data about the dental and periodontal care of cardiac patients with PD were asked, and all data were presented as descriptive percentages and then analyzed using the Kappa test. *Results*: From a total of 153 (100%) DUS, the sample was mostly composed of 104 (68%) female participants, with an average age of 21.1 years. Regarding basic knowledge, the majority of answers were no, with G1 being higher than G2 and G3. Regarding clinical questions, 1247 (58.3%) answers were no. Additionally, regarding fundamental clinical questions 1, 2, 3, 7, 9, 11, 13 and 14, the majority of G1, G2 and G3 answered no, demonstrating a major lack of knowledge. *Conclusions*: In our study, DUS demonstrated a low knowledge level of the dental and periodontal care of cardiac patients with PD and its bi-directional link. Thus, according to our results, an improvement in dentistry educational programs regarding periodontal medicine must be implemented.

## 1. Introduction

Periodontal disease (PD) is established as a multifactorial, chronic infectious-inflammatory non-communicable disease (NCD), mainly caused by dental plaque accumulation associated with poor oral hygiene leading to dysbiosis. If it remains untreated or incorrectly treated, PD will cause several non-reversible damages to the alveolar bone and supportive tissues such as periodontal ligament and cementum, a major consequence of which is the clinical attachment loss (CAL) of supportive tissue, resulting in teeth loss [1,2,3,4]. Additionally, since 2018, according to Caton et al. [5], PD is currently classified as healthy gums, gingivitis or periodontitis and PD associated with systemic diseases.

From the onset of dysbiosis. an increased concentration of periodontal pathogenic bacteria such as *P. intermedia*, *F. nucleatum*, *P. gingivalis*, *T. denticola* and *T. forsythia* are present within the dental plaque, releasing some enzymes, such as lipopolysaccharides and cytotoxins, that activate the host’s immune response [6,7,8]. This is followed by a slowly increasing augmentation of proinflammatory mediators and cytokines, such as matrix metalloproteinases (MMP), tumor necrosis factor-alpha (TNF-α) and osteoclasts that participate in extracellular matrix remodeling and bone destruction, which aggravate CAL and periodontal pocket formation and growth [9,10,11,12], creating a subgingival microbiome known as a “bacterial pool” [13,14,15].

Eventually, through this etiopathogenesis mechanism linked with host immunosuppression, recent studies have demonstrated that periodontal pathogens can damage tissue epithelium and vascular endothelium from the periodontal pocket to the bloodstream, causing bacterial translocation throughout different systemic tissues. This develops an intense systemic proinflammatory response, which probably affects and even increases the severity of pathological conditions such as diabetes and cardiovascular disease (CVD) [16,17,18]. In the last two decades, some studies have strongly demonstrated the innumerable correlations between a number of diseases or systemic conditions and PD and tried to describe how periodontal infectious-inflammatory disease connects to other systemic NCD. Thus, a new research field known as periodontal medicine has been created [19].

The World Health Organization (WHO) stated that CVD is a general medical term for a group of systemic disorders affecting heart or blood vessels, composed of systemic arterial hypertension (SAH), coronary artery disease, cerebrovascular accident or stroke, peripheral arterial disease, rheumatic heart disease, congenital cardiomyopathies, arterial thrombosis, deep vein thrombosis and pulmonary thromboembolism [20]. The main immunopathogenesis mechanism of CVDs is the accumulation of cholesterol debris or adhesion of adipose tissue in the vascular endothelium, causing an intense secretion of proinflammatory mediators such as TNF-α, cytokines such as interleukin-6 (IL-6), C-reactive protein (CRP) and MMP and leukocytes, creating a major local and systemic proinflammatory reaction [21].

A common association between PD and CVD is seen in the literature between periodontitis and atherosclerosis or SAH. Nevertheless, although several studies have determined a clinical correlation, a detailed understanding between these two pathologies has not been clarified yet. However, over the years, a few proposed mechanisms and theories have been established to explain the association between PD and CVD: endothelial dysfunction, mutual mediators of inflammation and periodontal pathogen-related bacteremia [22,23,24,25,26,27,28,29,30,31]. Therefore, for this reason, the correct periodontal treatment and the multidisciplinary follow-up of patients with cardiac disease and PD is of great importance. However, a few studies have indicated that dentists’ knowledge regarding the correct treatment and management of cardiac patients with PD is not as suitable as it should be, and this lack of knowledge is not yet described among dentists and, mainly, dental students, who are the future of dental management [32,33,34,35].

Therefore, teaching the importance of periodontal medicine and the association between oral health and systemic disease association during dentistry education will certainly improve the quality of students and future dentists. The role of dentistry has changed from the diagnosis and treatment of only carious teeth to oral health promotion and the knowledgeable care of patients with an association between oral and systemic diseases. Unfortunately, only a few studies have attempted to demonstrate the effects of this knowledgeable care. As stated by Faden et al. [35] and Fonseca et al. [36], these few studies evidence that knowledge of the association between oral and systemic diseases is not well established among dentists, while dentistry undergraduate students (DUS) have been neglected by these studies, resulting in a need to fill this gap in the literature. Therefore, this present study aimed to determine and evaluate the knowledge level, doubts and difficulties of DUS regarding the correct periodontal treatment and management of cardiac patients with PD.

## 2. Materials and Methods

### 2.1. Study Characterization and Sample Size

This descriptive, population-based, cross-sectional study was conducted with undergraduates in dentistry at colleges in northern Brazil. Data collection occurred between March and June 2022 in a fully digital study through the Google Forms Platform (Google Inc., Mountain View, CA, USA). Participants were selected from DUS and divided into 3 groups according to their period/semester in dentistry education during the study time.

The groups were as follows: G1—students in their first period/semester in dentistry graduation who never had classes on periodontics and periodontal medicine linked to CVDs (control group); G2—students in their fifth period/semester in dentistry graduation who had one class on periodontics and periodontal medicine linked to CVDs but no clinical management of these type of patients; G3—students in their tenth period/semester in dentistry graduation who had at least two classes on periodontics and periodontal medicine linked to CVDs and had previous experience in the clinical management of these type of patients.

### 2.2. Ethics

All procedures performed in this study involving participants were conducted in accordance with institutional and/or national research committee ethical standards and the 1964 Declaration of Helsinki and its subsequent amendments or comparable ethical standards. This study was approved by the Research Ethics Committee of the Institute of Biological Sciences at the Federal University of Pará under protocol number 5.190.098, and informed consent was obtained from all participants involved in this study.

### 2.3. Data Collection

A non-probability “snowball” sampling technique was used to access and invite dentistry students to participate in this study. Initially, three students known in the selected community (class leaders) were directly contacted by the study authors. They received information about the objectives, data collection and the importance of conducting this study. Then, these three students began to publicize the study and to invite, through social media, other students from different dental colleges in northern Brazil to compose the full sample size.

The selected apps for communication among the participants were WhatsApp (Facebook Inc, Menlo Park, CA, USA), Telegram (Telegram Messenger LLP, Moscow, Russia), Instagram (Facebook Inc, Menlo Park, CA, USA) and Facebook (Facebook Inc, Menlo Park, CA, USA). All participants received a link to access the digital form. Later, each student signed a digital written consent form and then agreed to participate in the study. After completing and sending the information through the form, each student also invited three other students to fill out the digital form, and this procedure was repeated several times in order to compose the full sample size [36].

All participants were informed about the study’s nature, objective, potential risks and benefits, and if they were in agreement with all, they were invited to sign a digital written consent form. The inclusion criteria for this study were: age ≥ 18 years old, dentistry undergraduate students, Pará state resident, having access to the internet and being in the selected period/semester in dentistry graduation. All potential participants who did not sign a digital written consent form or did not meet inclusion criteria were excluded from the study.

### 2.4. Digital Form

This study used a digital form as a data collection instrument. This instrument was formatted and administered through the Google Forms Platform (Google Inc., Mountain View, CA, USA) and monitored by the authors. Its distribution took place through an electronic link directly sent to all participants. The digital form was composed of four blocks: the first block contained digital research information; the second block contained consent terms, and the third and fourth blocks contained 25 queries about socioepidemiological profiles and clinical-theoretical knowledge. Participants could not proceed to the next block without providing the required information on the current block. Block 3 contained 12 questions about the socioepidemiological characteristics, and block 4 contained 13 queries about various clinical-theoretical knowledge related to the correct periodontal treatment and management of cardiac patients with PD.

### 2.5. Statistical Analysis

All data from this study were entered into an Excel database (Microsoft Corp., Redmond, WA, USA) and converted to BioEstat. Statistical parameters (absolute and relative frequencies, mean, median, range and standard deviation) were used to describe the characteristics of the sample according to the quantitative and qualitative variables investigated [36].

All 25 questions were organized into two models: (a) questions with dichotomous answers and (b) questions with multiple-choice answers. Data were evaluated by absolute and relative frequency, as well as by G tests, which were used to compare informed knowledge and knowledge demonstrated by the students. The value of *p* < 0.005 was considered significant for all analyses, and all statistical procedures were carried out in the BioEstat 5.0 software [36]. Additionally, in the selected groups, an associated 95% confidence interval (CI) was used to measure the strength of the dependent association between yes and no answers in query block 4.

## 3. Results

### 3.1. Sample, Socioepidemiological General Knowledge Characteristics

From a total of 200 DUS who were contacted following the invitation of the original three DUS, 153 (76.5%) participated in the present study. Of those 200 (100%), 47 (23.5%) were not included in the analyses because they did not meet the inclusion criteria. These individuals not included did not sign the informed term of consent or did not fill the digital form correctly. All socioepidemiological characteristics of the 153 DUS are shown in Table 1 (block 3). The sample was composed of G1 (n = 48–31.4%), G2 (n = 53–34.6%) and G3 (n = 52–34%) students. Regarding gender, female DUS (n = 104–68%) were more prevalent than male (n = 49–32%), and the G1 group had the most female individuals with 39 (81.2%). The mean age of the participants was 21.1 years (range: 18–35 years). Concerning college type, private colleges (n = 124–81%) were more prevalent than public colleges (n = 29–19%).

Table 1 also shows some important general knowledge of DUS about CVDs and PD. The first query was about how to measure blood pressure; 102 (66.7%) answered yes, with G3 (n = 50–96.1%) representing almost half of all affirmative answers; G1 (n = 35–73%) had the most negative answers. Regarding blood pressure parameters, 104 (68%) answered yes, with G3 (n = 47–90.3%) having the most positive answers and G1 (n = 44–91.6%) having the most negative answers. About the use of digital or/and manual blood pressure measuring devices, 120 (78.4%) answered yes, with G3 (n = 50–96.1%) having the most positive answers. Regarding having witnessed any kind of cardiac incident, 144 (94.1%) individuals answered yes, although the next question was if the DUS knew how to prevent any kind of cardiac incident. Unfortunately, 144 (94.1%) answered no.

The next question was about any kind of knowledge of interactions between local dental anesthetics and antihypertensive drugs. A total of 84 students (55%) said no. Unexpectedly, G2 (n = 24–45.3%) and G3 (n = 25–48%) had increased numbers of negative answers, despite being at a period/semester in dentistry graduation where they had classes about interactions between CVD and PD, as well as clinical experience with these topics. Regarding if they had already provided any oral care to cardiac patients with PD, 96 (62.8%) said no. Regarding having received any content about interactions between CVD and PD, the answers were unusually divided because G2 and G3 had, in their educational programs, classes about interactions between CVD and PD. Thus, 77 (50.3%) answered yes, and 76 (49.7%) gave a negative answer. Finally, it was asked if individuals had any doubts or apprehensions about the management of cardiac patients with PD, to which 91 (59.4%) answered yes, with statistics between G1 (n = 41–85.4%), G2 (n = 40–75.4%) and G3 (n = 35–67.3%) being almost even.

A significant association was found between age, clinically important knowledge questions (knowing how to measure blood pressure, knowing about blood pressure parameters, knowing how to use digital and manual blood pressure measuring devices) and previous access to any content about cardiac patients with PD, which demonstrates the importance of cardiology in dental treatment.

### 3.2. Self-Reported Technical Knowledge about Interactions between CVD and PD

The answers to technical questions on topics regarding interactions between CVD and PD (block 4) are summarized in Table 2. The questions in block 4 were dichotomous questions to evaluate individuals’ knowledge about interactions between CVD and PD. The majority of individuals answered no to questions 1, 2, 3, 7, 9, 11, 13 and 14, in a total of 2.142, with 895 (41.7%) answering yes and 1247 (58.3%) answering no. On block 4, the first query was if DUS have any knowledge about interactions between PD and CVD and their consequences; 112 (73.2%) answered no. The next query was if they can make a correct periodontal treatment plan for cardiac patients, and 120 (79.4%) answered no. The next query was about their individual knowledge of how interactions between PD and CVD happen, and 102 (66.7%) answered no.

Another query was about if individuals know how to examine cardiac patients with PD, to which 88 (57.5%) answered yes. The following queries 5, 6, 7 and 8 were about periodontal health parameters (n = 118–77.1%) and PD parameters (n = 98–64.1%), and were both yes in their majority. However, the next queries were about common signs and symptoms of periodontal health and PD, respectively, in cardiac patients. These queries were set to confirm or deny the students’ previously self-declared knowledge about periodontal health parameters and PD parameters. Regarding common signs and symptoms of periodontal health, 90 (58.9%) answered no and 77 (50.3%) answered yes about common signs and symptoms of PD parameters, although the answers on common signs and symptoms of PD parameters had nearly similar percentages.

Concerning whether DUS knew how to interpret a resting electrocardiogram, 142 (92.8%) answered no, showing a lack of knowledge regarding this topic in dentistry education and formation because resting electrocardiogram is an important preoperative laboratory exam, which can detect cardiac abnormalities such as arrhythmias, SAH or coronary heart disease. When asked about the importance of multidisciplinary treatment between dentists and cardiologists, 80 (52.3%) answered yes to knowing how important it is to improve treatment results. Regarding the main antihypertensive drugs used by cardiac patients in Brazil, 112 (73.2%) answered no, which is a major risk of drug or anesthetic interactions during periodontal treatment. Finally, 118 (77.1%) answered no to knowing how to provide correct oral care to cardiac patients with PD.

In Table 3, answers from Table 2 are presented by each group. In addition, a significant association between the 3 groups and specific periodontal clinical knowledge was found mainly in the following queries: knowing how to work with PD and CVD interactions, knowing how to examine cardiac patients with PD, knowing what PD parameters are, common signs and symptoms of periodontal health in cardiac patients, common signs and symptoms of PD in cardiac patients and knowing the importance of multidisciplinary treatment with cardiologistss.

## 4. Discussion

The present study aimed to identify the knowledge level, doubts and difficulties of DUS who attended dental colleges in northern Brazil during the dental management of cardiac patients. To the best of the author’s knowledge, which was obtained during a literature search, this study is the first to address the issue of the dental management of cardiac patients with PD among DUS.

Normally, the majority of studies try to understand the relationship between CVD and PD, but few studies have tried to understand the knowledge level, doubts and difficulties of DUS during dental management of cardiac patients with PD [32,33,34,35,36]. In this study, as mentioned before, our focus was the knowledge of DUS regarding cardiac patients with PD, and the main goal was to develop periodontal treatment for this specific population. Through a digital form, individuals reported their knowledge acquired along dentistry education, and, as observed, students in the 10^o^ period/semester in dentistry education, who were almost graduated and with previous clinical experience, demonstrated more knowledge than students in their 1^o^ period/semester in dentistry education. However, the answers provided clearly showed that most participants have doubts regarding the relationship between CVD and PD, as well as how to treat it correctly. Therefore, this can interfere in the clinical resolution of periodontal cases in cardiac patients.

In the early 2000s, Williams and Offenbacher [37] proposed new guidance for periodontology, and in their proposal, this new study field was titled periodontal medicine. Since then, several studies have tried to find the innumerable relationships between PD or periodontal conditions and systemic diseases such as CVD and diabetes. Most interactions occur through the presence of periodontal pathogens such as *P. gingivalis* and other periodontal microbiota in the systemic pathways, which impair hosts’ innate immunity defenses to develop an intense response and a more destructive host-microbial interplay in the periodontium. However, recent studies have demonstrated through biological and proinflammatory analyses of TNF-α, IL-6, CRP and MMP that this cytokine storm condition directly influences systemic conditions or diseases, such as diabetes and CVD.

Although periodontal medicine might not be a new term to periodontists worldwide, the general dentist population or DUS may not know about the importance of periodontal medicine in daily dentistry treatment. According to Faden et al. [35], the majority of their participants did not know the importance of periodontal medicine in the overall dentistry treatment. The authors suggest that the knowledge level among dentists may be raised easily by revising college curriculums and focusing on important topics about oral-systemic relationships. This study corroborates our results, from which, as shown in Table 2, it can be seen that out of a total of 2.142 responses, surprisingly, 1247 (58.3%) answers were negative regarding knowledge of common topics of periodontal medicine, mainly clinical questions 1, 2, 3, 7, 9, 11, 13 and 14 and mostly among G3, the test group with previous clinical experience and periodontal classes.

Unlike our results, Paquette et al. [34] evidenced that 75% of the individuals in their study had knowledge about oral systemic health or disease and risk factors; also, nearly 90% agreed that multidisciplinary medical-dental treatment was important. In a similar study, Bell et al. [33] demonstrated that 77% of dental hygienists in North Carolina presented a high knowledge level of oral-systemic evidence and interprofessional teamwork. In our results, G1 (n = 38–52%) demonstrated low knowledge about the importance of multidisciplinary medical-dental treatment; however, G3 (n = 35–31.3%) demonstrated a medium level of knowledge about the topic. This group should present a higher prevalence due to their previous clinical experience and periodontal classes.

To demonstrate the alarmingly low knowledge level (58.3%) among DUS in our study, Al-Mohaissen et al. [38] evaluated the knowledge level of 282 dentists regarding cardiac patients and their dental care, of whom 45.5% perceived cardiac patients as difficult to manage, although an interesting data point was that almost 90% dentist wished to receive education regarding the dental management of cardiac patients. In our study, there was not a query about receiving education about dental management in cardiac patients, although 112/153 (73.2%) answered negatively to a question asking if they have any knowledge about interactions between PD and CVD and their consequences, and 120/153 (79.4%) answered that they did not know how to plan a correct periodontal plan of treatment for cardiac patients. We can therefore infer by our results that more educational classes about dental management in cardiac patients might improve DUS’ knowledge level.

Mohideen et al. [39] evaluated the knowledge of dental students about medical emergencies and management. The majority of the authors’ queries were about cardiac parameters, such as knowledge about antihypertensive drugs or CVD drug interactions with local dental anesthetics such as lidocaine. Mohideen et al. [39] stated that most dental students were mindful regarding vital signs, and 27–37% of the respondents knew to identify angina, transient ischemia, and lidocaine toxicity. Therefore, based on their results, the authors inferred that dental students needed more training about cardiac patients before graduation. In our results, 112/153 (73.2%) DUS did not know about the main antihypertensive drugs used by cardiac patients in Brazil, which could compromise dental treatment and outcomes in a medical emergency. Furthermore, 84/153 (55%) DUS claimed not to know about interactions between cardiac drugs and local dental anesthetics such as as lidocaine, and 91/153 (59.4%) DUS have doubts or apprehensions about the management of cardiac patients with PD, results that corroborate the study by Mohideen et al. [39].

Perhaps, the COVID-19 pandemic might demonstrate that using different educational technologies such as robotics, digital classes, forms, e-books or artificial intelligence can improve the dentistry educational process and enhance dentists’ or dental students’ knowledge of various topics such as the dental management of cardiac patients with PD. However, some discussion about educational technologies in higher education must be made mainly because in developing countries or regions such as the northern region in Brazil, access to these technologies may be scarce. Analyzing our results, this study was successful in terms of data collection and identifying the main doubts of the individuals, although we had certain limitations, such as the relatively small sample size, demographic restriction to the Pará state, the inclusion of only undergraduates in dentistry from northern Brazil colleges, as well as the involuntary exclusion of individuals who did not have access to the internet [40,41,42].

Therefore, as a recommendation, perhaps in the future, a further study must be carried out by contacting all past students who participated in this study to ask them to complete another form to see if, after graduation, their knowledge improved as dentists or even periodontists. Additionally, new technologies such as e-books or video classes might be inserted in dentistry college programs, as well as more classes regarding periodontal medicine and the importance of oral-systemic health knowledge among future dentists worldwide.

## 5. Conclusions

In conclusion, our DUS, in general, demonstrated a low knowledge level or awareness regarding the relationship between CVD and PD despite their specific classes and clinical experiences during dentistry education. Additionally, the prevalence of doubts about drug interactions, laboratory exam interpretation, how to make a treatment plan and the importance of multidisciplinary treatment increased among our students. Therefore, we need to comprehend how dentistry educational programs’ methodologies are teaching the relationship between oral and systemic diseases and improve or even change these methods in order to promote a better understanding of the oral-systemic disease relationship, especially between CVDs and PD. Didactic material development might also simprove this knowledge.

## Figures and Tables

**Table 1 medicina-59-00509-t001:** Socioepidemiological data and general knowledge about cardiac patient management of DUS in 3 different periods/semesters in dentistry graduation.

Parameters	Total	1^o^ Period/Semester (G1)	5^o^ Period/Semester (G2)	10^o^ Period/Semester (G3)	*p*-Value ^†^
Total	153 (100%)	48 (31.4%)	53 (34.6%)	52 (34%)	
Gender *					
Male	49 (32%)	9 (18.7%)	18 (34%)	22 (42.3%)	0.0355
Female	104 (68%)	39 (81.2%)	35 (66%)	30 (57.7%)	
Age (years) **					
18–23	55 (36%)	42 (87.5%)	5 (9.4%)	8 (15.4%)	<0.0001
24–29	93 (60.7 %)	6 (12.5%)	47 (88.6 %)	40 (77%)	
30–35	5 (3.3%)	-	1 (2%)	4 (7.6%)	
College type *					
Private	124 (81%)	40 (83.3%)	37 (70%)	47 (90.3%)	0.0259
Public	29 (19%)	8 (16.7%)	16 (30%)	5 (9.7%)	
Know how to measure blood pressure **					
Yes	102 (66.7%)	2 (4%)	38 (71.7%)	50 (96.1%)	<0.0001
No	19 (10.5%)	35 (73%)	3 (5.7%)	-	
Maybe	30 (19.6%)	11 (23%)	12 (22.6%)	2 (3.9%)	
Know about blood pressure parameters **					
Yes	104 (68%)	1 (2.2%)	3 (5.7%)	47 (90.3%)	<0.0001
No	19 (12.4%)	44 (91.6%)	6 (11.3%)	1 (2%)	
Maybe	42 (40%)	3 (6.2%)	44 (83 %)	4 (7.7%)	
Know how to use digital and manual blood pressure measuring devices **					
Yes	120 (78.4%)	33 (68.7%)	37 (70%)	50 (96.1%)	<0.0001
No	13 (8.5%)	12 (25%)	1 (1.7 %)	-	
Maybe	20 (13.1%)	3 (6.3%)	15 (28.3%)	2 (3.9%)	
Have you ever witnessed any kind of cardiac patient with PD incident *					
Yes	9 (5.8%)	2 (4.2%)	3 (5.7%)	4 (7.7%)	0.7672
No	144 (94.1%)	46 (95.8%)	50 (94.3%)	48 (92.3%)	
Know how to prevent incidents of cardiac patient with PD *					
Yes	6 (4%)	1 (2%)	2 (3.8%)	3 (5.8%)	0.6573
No	147 (96%)	47 (98%)	51 (96.2%)	49 (94.2%)	
Have any knowledge about CVDs drug interactions to dental local anesthetics *					
Yes	69 (45%)	13 (27 %)	29 (54.7%)	27 (52%)	0.009
No	84 (55%)	35 (73%)	24 (45.3%)	25 (48%)	
Already provided oral care to cardiac patient with PD *					
Yes	57 (37.2%)	10 (20.9%)	23 (43.4%)	24 (46.2%)	0.0144
No	96 (62.8%)	38 (79.1%)	30 (56.6%)	28 (53.8%)	
Had access to any content (class or video or paper) about cardiac patient with PD *					
Yes	77 (50.3%)	8 (16.7%)	36 (68%)	34 (65.3%)	<0.0001
No	76 (49.7%)	40 (83.3%)	17 (32%)	18 (34.7%)	
Have doubts or apprehensions about cardiac patient with PD management *					
Yes	91 (59.4%)	41 (85.4%)	40 (75.4%)	35 (67.3%)	0.1039
No	62 (40.6%)	7 (14.6%)	13 (24.6%)	17 (32.7%)	

* Dichotomous; ** multiple-choice questions; ^†^ G test.

**Table 2 medicina-59-00509-t002:** Technical knowledge of DUS about cardiac patient management.

Questions	Answers			
	Yes n = 895 (41.7%)	95% CI	No n = 1247 (58.3%)	95% CI
1. Have any knowledge about PD and CVD interactions and consequences	41 (26.8%)	0.198 (19.8%)–0.338 (33.8%)	112 (73.2%)	0.662 (66.2%)–0.802 (80.2%)
2. Know how to plan the correct periodontal treatment of cardiac patients	33 (21.6%)	0.151 (15.1%)–0.281 (28.1%)	120 (79.4%)	0.719 (71.9%)–0.849 (84.9%)
3. Know how to work PD and CVD interactions	51 (33.3%)	0.259 (25.9%)–0.408 (40.8%)	102 (66.7%)	0.592 (59.2%)–0.741 (74.1%)
4. Know how to examine cardiac patients with PD	88 (57.5%)	0.497 (49.7%)–0.653 (65.3%)	65 (42.5%)	0.347 (34.7%)–0.503 (50.3%)
5. Know what periodontal health parameters are	118 (77.1%)	0.705 (70.5%)–0.838 (83.8%)	35 (22.9%)	0.162 (16.2%)–0.295 (29.5%)
6. Know what PD parameters are	98 (64.1%)	0.564 (56.4%)–0.717 (71.7%)	55 (35.9%)	0.283 (28.3%)–0.436 (43.6%)
7. Common signs and symptoms of periodontal health in cardiac patients	63 (41.1%)	0.334 (33.4%)–0.490 (49.0%)	90 (58.9%)	0.510 (51.0%)–0.666 (66.6%)
8. Common signs and symptoms of PD in cardiac patients	77 (50.3%)	0.424 (42.4%)–0.582 (58.2%)	76 (49.7%)	0.418 (41.8%)–0.576 (57.6%)
9. Know what correct laboratory blood test values are	110 (71.9%)	0.648 (64.8%)–0.790 (79.0%)	43 (28.1%)	0.210 (21.0%)–0.352 (35.2%)
10. Know how to interpret a resting electrocardiogram	11 (7.2%)	0.031 (3.1%)–0.113 (11.3%)	142 (92.8%)	0.887 (88.7%)–0.969 (96.9%)
11. Know the importance of multidisciplinary treatment with cardiologists	80 (52.3%)	0.444 (44.4%)–0.602 (60.2%)	73 (47.7%)	0.398 (39.8%)–0.556 (55.6%)
12. What are the main antihypertensive drugs of cardiac patients	41 (26.8%)	0.198 (19.8%)–0.338 (33.8%)	112 (73.2%)	0.662 (66.2%)–0.802 (80.2%)
13. Know how to provide correct oral care to cardiac patients with PD	35 (22.9%)	0.162 (16.2%)–0.295 (29.5%)	118 (77.1%)	0.705 (70.5%)–0.838 (83.8%)

**Table 3 medicina-59-00509-t003:** Technical knowledge of DUS about cardiac patient management through each group.

Questions	1^o^ Period/Semester (G1) n = 48		5^o^ Period/Semester (G2) n = 53		10^o^ Period/Semester (G3) n = 52		*p*-Value ^†^
	Yes n = (%)	No n = (%)	Yes n = (%)	No n = (%)	Yes n = (%)	No n = (%)	
Have any knowledge about PD and CVD interactions and consequences	5 (12.2%)	43 (38.4%)	16 (39%)	37 (33%)	20 (48.8%)	32 (28.6%)	0.0034
Know how to plan the correct periodontal treatment of cardiac patients	2 (6.1%)	46 (38.3%)	14 (42.4%)	39 (32.5%)	17 (51.5%)	35 (29.2%)	0.0004
Know how to work PD and CVD interactions	5 (9.8%)	43 (42.1%)	26 (51%)	27 (26.6%)	20 (39.2%)	32 (31.3%)	<0.0001
Know how to examine cardiac patients with PD	12 (13.6%)	36 (55.3%)	32 (36.3%)	21 (32.3%)	44 (50.1%)	8 (12.4%)	<0.0001
Know what periodontal health parameters are	32 (27.1%)	16 (45.7%)	41 (34.7%)	12 (34.2%)	45 (38.2%)	7 (20.1%)	0.0623
Know what PD parameters are	16 (16.3%)	32 (58.1%)	37 (37.7%)	16 (29.1%)	45 (46%)	7 (12.8%)	<0.0001
Common signs and symptoms of periodontal health in cardiac patients	3 (4.7%)	45 (50%)	24 (33.3%)	29 (32.2%)	39 (62%)	13 (17.8%)	<0.0001
Common signs and symptoms of PD in cardiac patients	5 (6.5%)	43 (56.5%)	37 (48%)	16 (21%)	35 (45.5%)	17 (22.5%)	<0.0001
Know what correct laboratory blood test values are	32 (29.1%)	16 (37.2%)	43 (39.1%)	10 (23.2%)	35 (31.8%)	17 (39.6%)	0.1723
Know how to interpret a resting electrocardiogram	1 (9.1%)	47 (33.1%)	4 (36.3%)	49 (34.5%)	6 (54.6%)	46 (32.4%)	0.1646
Know the importance of multidisciplinary treatment with cardiologists	10 (12.5%)	38 (52%)	45 (56.2%)	8 (11%)	35 (31.3%)	17 (37%)	<0.0001
What are the main antihypertensive drugs of cardiac patients	4 (9.7%)	44 (39.2%)	20 (48.7%)	33 (29.4%)	17 (41.6%)	35 (31.4%)	0.0009
Know how to provide correct oral care to cardiac patients with PD	12 (34.2%)	36 (30.5%)	11 (31.4%)	42 (35.6%)	12 (34.4%)	40 (33.9%)	0.8805

^†^ G test.

## Data Availability

Not applicable.

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
