# Peer review of "Digital Form for Assessing Dentistry Undergraduates Regarding Periodontal Disease Associated with Cardiovascular Diseases"

_medicina, 2023, doi:10.3390/medicina59030509_

Round 1

Reviewer 1 Report

Dear Authors,

After reading the title of your article, I was expecting to find some correlation with the COVID pandemic. At least regarding the stress level of the dental students. Or comparing students before, during and after pandemic. Thus, the words "during COVID-19 pandemic" have no reason to be in the title.

Another problem that is not mentioned is that these students are during the education process, we cannot expect them to know too much about periodontal disease and cardiovascular diseases. It is a period in which they acquire information and learn to make connections between various disciplines. This evaluation would have been optimal for periodontologists or dentists distributed during the practice period.

The article is suited for a section regarding dental education and not dentistry. It aims to determine the doubts and difficulties of dentistry undergraduates students.

The elaboration of the article and the quality of the statistical study are very good. The results are well presented

What is the use of reference 36 in lines 121,129, 159 where the data collection is described?

I recommend a minor revision.

Author Response

Reply to reviewer #1

1. Concern of the reviewer

• After reading the title of your article, I was expecting to find some correlation with the COVID pandemic. At least regarding the stress level of the dental students. Or comparing students before, during and after pandemic. Thus, the words "during COVID-19 pandemic" have no reason to be in the title. 

Our response: Dear Reviewer #1, we appreciate your suggestion and the tittle were carefully revised, we added this part because the study was made during COVID-19. 

Revised text:Page 1, lines 2-3, Digital form for assessing dentistry undergraduates about periodontal disease associated with cardiovascular diseases.” 

2. Concern of the reviewer

• Another problem that is not mentioned is that these students are during the education process, we cannot expect them to know too much about periodontal disease and cardiovascular diseases. It is a period in which they acquire information and learn to make connections between various disciplines. This evaluation would have been optimal for periodontologists or dentists distributed during the practice period. 

Our response: Dear Reviewer #1, we appreciate your concern and the text was carefully explained to readers. 

Revised text:Page 2, lines 89-92, “so among dentists and, mainly, dental students which are the future of dental management [32-35]. So, to teach the importance of periodontal medicine and oral health-systemic disease association during dentistry graduation certainly will improve the quality of students and future dentists.”

Page 9, lines 304-307, “so, as recommendations, perhaps in prior future a next study must be made contacting all past students who participated of this study and then ask them another form to see if after graduation their knowledge improved as dentists or even periodontists” 

3. Concern of the reviewer

• The article is suited for a section regarding dental education and not dentistry. It aims to determine the doubts and difficulties of dentistry undergraduates students. The elaboration of the article and the quality of the statistical study are very good. The results are well presented. 

Our response: Dear Reviewer #1, we appreciate your kind words and would be very grateful if you recommend this paper for publication in this section in order to contribute to all dentists to improve our quality in management of cardiac patients. 

4. Concern of the reviewer

• What is the use of reference 36 in lines 121,129, 159 where the data collection is described? 

Our response: Dear Reviewer #1, we appreciate your suggestion and the text was carefully revised, we used this reference as base to our snowball sampling technique.  

Revised text:Page 3, line 124, “other students from different dental colleges in northern Brazil to compose the full sample size.” 

5. Concern of the reviewer • I recommend a minor revision. 

Our response: Dear Reviewer #1, we appreciate your all suggestions and we tried our best to improve this paper according your insightful suggestions and we hope to achieve your recommendation to publication.

Reviewer 2 Report

Dear Authors,

Congratulations on your good study-

But I have a few concerns –

Rewrite the background of the abstract.

What is G1,G2...etc in the abstract?

The last line of the conclusion, in the abstract, is not properly written – rewrite it.

Line 87-88-

Describe the previous studies which show lacunae in the field of research then put how your study is going to fill that lacuna….then write the aim….

Line 98-99-

What do you mean by-semester?

– explain it as different nomenclature is used in various parts of the world to describe the student level.

What were the validity and reliability of the questions?

Line 248-260-

Rewrite this ,,,, as your aim was different …what you have mentioned in these lines may be your objective ….

Discussion of the study is very short add ore discussion- write about your results compare it with the previous study, determine the inferences from your results discuss it …..

Write limitations and recommendations as separate paragraphs – this will enhance your study quality…

Write in your recommendations- what you suggest for improving the knowledge of students. why you selected the dental students and how it will be helpful in future …..

Suggest you check the possibility of adding a few recent references along with others, which may be useful in modifying your paper in the discussion section more appropriately. –

1.     Predoctoral dental implant education techniques-students' perception and attitude. J Dent Educ. 2021;85(3):392-400. doi:10.1002/jdd.12453

2.     Role of Robotics and Artificial Intelligence in Oral Health and Preventive Dentistry - Knowledge, Perception and Attitude of Dentists. Oral Health Prev Dent. 2021;19(1):353-363. doi:10.3290/j.ohpd.b1693873

3.     Perception about Health Applications (Apps) in Smartphones towards Telemedicine during COVID-19: A Cross-Sectional Study. J. Pers. Med. 202212, 1920. doi: 10.3390/jpm12111920

Author Response

Reply to reviewer #2

1. Concern of the reviewer             

• Rewrite the background of the abstract. What is G1,G2...etc in the abstract? The last line of the conclusion, in the abstract, is not properly written – rewrite it. 

Our response: Dear Reviewer #2, we appreciate your concern and the text was carefully revised.  

Revised text:Page 1, lines 20-42, “Background: Throughout recent years periodontal disease (PD) has been linked to innumerous medical systemic conditions like cardiovascular diseases (CVDs), which this association could impact negatively oral health, so the knowledge of dentists, since graduation, needs to follow modern dentistry in order to promote oral health, mainly, in systemic compromised patients. Therefore the present study aimed to determine and evaluate the knowledge level of dentistry undergraduates students (DUS) regarding correct periodontal treatment and management of cardiac patients with PD. Methods: This cross-sectional and populational-based study was conducted between March and June 2022, in the northern Brazil, a total of 153 DUS received an anonymous digital form (Google® Forms Platform) using a non-probabilistic “snowball” sampling technique, the digital form was composed of four blocks dichotomous and multiple-choice questions. After signed the informed consent term, DUS were divided into three groups according their period/semester in dentistry graduation during the study time (G1: 1st period/semester; G2: 5th period/semester and G3: 10th period/semester). A total of 25 questions referred to demographic, educational and knowledge data about dental and periodontal care of cardiac patients with PD were asked and all data were presented as descriptive percentages and then analyzed using the Kappa test. Results: From total of 153 (100%) DUS, the sample was mostly composed of 104 (68%) female participants, with an average age of 21.1 years. Regarding basic knowledge the majority of answers were no with G1 being higher than G2 and G3, about clinical questions 1247 (58.3%) answers were no, also fundamental clinical questions 1, 2, 3, 7, 9, 11, 13 and 14 the majority of G1, G2 and G3 answered no demonstrating a major lack of knowledge. Conclusions: In our study, DUS demonstrated to have a low knowledge level dental and periodontal care of cardiac patients with PD and its bi-directional link, so according our results an improvement in dentistry educational programs about periodontal medicine must be implemented.” 

2. Concern of the reviewer• Line 87-88 - Describe the previous studies which show lacunae in the field of research then put how your study is going to fill that lacuna….then write the aim…. 

Our response: Dear Reviewer #2, we appreciate your concern. The text was carefully added.

Revised text:Pages 2-3, lines 97-103, “The role of dentistry changed from the diagnosis and treatment of only carious teeth to oral health promotion and knowledge care of patients with oral-systemic diseases association, unfortunately just few studies try to demonstrate if this knowledge care and as stated by Faden et al.[35] and Fonseca et al.[36] these few studies evidence that oral-systemic diseases association knowledge is not well established among dentists, while dentistry undergraduate students (DUS) has been neglected by these studies, so there is a need to fill this gap in literature.”

3. Concern of the reviewer

Line 98-99 - What do you mean by-semester? – explain it as different nomenclature is used in various parts of the world to describe the student level.

Our response: Dear Reviewer #2, we appreciate your suggestion. The text was carefully explained and where this lack of explanation as presented it was rewritten. 

Revised text:Page 3, lines 112-119,their period/semester in dentistry graduation during the study time. The groups were: G1 –first period/semester in dentistry graduation, never had classes on periodontics and periodontal medicine linked to CVDs (control group); G2 –fifth period/semester in dentistry graduation, which had 1 class on periodontics and periodontal medicine linked to CVDs, but no clinical management of these type of patients; G3 –tenth period/semester in dentistry graduation, who had at least 2 classes on periodontics and periodontal medicine linked to CVDs and had previous experience on clinical management of these type of patients.”

4. Concern of the reviewer

What were the validity and reliability of the questions?

Our response: Dear Reviewer #2, we appreciate your concern. The queries were based on the references (Faden, A.A.; Alsalhani, A.B.; Idrees, M.M.; Alshehri, M.A.; Nassani, M.Z.; Kujan, O.B. Knowledge, attitudes, and practice behavior of dental hygienists regarding the impact of systemic diseases on oral health. Saudi Med J 2018, 39(11):1139-1147./ de Souza Fonseca, R.R.; Laurentino, R.V.; de Menezes, S.A.F.; Oliveira-Filho, A.B.; Alves, A.C.B.A.; Frade, P.C.R.; Machado, L.F.A. Digital Form for Assessing Dentists' Knowledge about Oral Care of People Living with HIV. Int J Environ Res Public Health 2022, 19(9):5055.) and also based in a non-published pilot study. 

5. Concern of the reviewer

Line 248-260 - Rewrite this ,,,, as your aim was different …what you have mentioned in these lines may be your objective ….

Our response: Dear Reviewer #2, we appreciate your suggestion. The text was carefully rewritten. 

Revised text:Page 9, lines 260-267,The present study aimed to identify knowledge level, doubts and difficulties of DUS during dental management of cardiac patients with PD attented at dental colleges in northern Brazil and to the best of author’s knowledge during literature searching this study is the first to address the issue of dental management of cardiac patients with PD among DUS. Normally, the majority of studies try to understand the relationship between CVDs and PD, but few studies tried to understand knowledge level, doubts and difficulties of DUS during dental management of cardiac patients with PD [32-36].”  

6. Concern of the reviewer

Discussion of the study is very short add ore discussion- write about your results compare it with the previous study, determine the inferences from your results discuss it

Our response: Dear Reviewer #2, we appreciate your suggestion. The text was carefully added. 

Revised text:Page 10, lines 305-334,To demonstrated alarming low knowledge level (58.3%) among DUS in our study, Al-Mohaissen et al. [38] evaluated knowledge level of 282 dentists about cardiac patients and their dental care, of whom 45.5% perceived cardiac patients as difficult to manage, although an interesting data was that almost 90% dentist wished to receive education regarding dental management in cardiac patient. In our study there was not a query about receive education about dental management in cardiac patient, although 112/153 (73.2%) answered no if they have any knowledge about PD and CVDs interactions and consequence and 120/153 (79.4%) answered no if they know how to plan a correct periodontal treatment to cardiac patients, so we can infer by our results that more educational classes about dental management in cardiac patient might improve DUS knowledge level.Mohideen et al. [39] evaluated the knowledge of dental students about medical emergencies and management, in authors queries the majority was about cardiac parameters, like knowledge about antihypertensive drugs or CVDs drug interactions to dental local anesthetics as lidocaine. Mohideen et al. [39] stated that most of dental students were mindful regarding vital signs, also 27%–37% of the respondents knew to identify angina, transient ischemia, and lidocaine toxicity, so based on their results authors inferred that dental students needed more training about cardiac patients before graduation. In our results 112/153 (73.2%) do not know about the main antihypertensive drugs used by cardiac patients in Brazil, which could compromise dental treatment and outcome in a medical emergency, also 84/153 (55%) DUS claimed not know about cardiac drugs interactions with dental local anesthetics as lidocaine and 91/153 (59.4%) DUS have doubts or apprehensions about cardiac patient with PD management, results that corroborates Mohideen et al. [39] study.  Perhaps, COVID-19 pandemic might evidenced that using different educational technologies such as robotics or digital class or forms or e-books or artificial intelligence to improve dentistry educational process and enhance dentists or dental students knowledge through various topics like dental management of cardiac patients with PD, although, some discussion about educational technologies in higher education must be made mainly because in developing countries or regions such as northern region in Brazil access to these technologies may be scarce.”  

7. Concern of the reviewer

Write limitations and recommendations as separate paragraphs – this will enhance your study quality.

Our response: Dear Reviewer #2, we appreciate your suggestion. The text was carefully separated. 

Revised text:Page 11, lines 334-339,Analyzing our results, this study was successful in terms of data collection and identifying the main doubts of the individuals, although we had certain limitations, such as the relatively small sample size, demographic restriction to Pará state, only undergraduates in dentistry from northern Brazil college, as well as there was involuntary exclusion of individuals who did not have access to the internet.”  

8. Concern of the reviewer

Write in your recommendations- what you suggest for improving the knowledge of students. why you selected the dental students and how it will be helpful in future.

Our response: Dear Reviewer #2, we appreciate your suggestion. The text was carefully added. 

Revised text:Page 11, lines 340-345,So, as recommendations, perhaps in prior future a next study must be made contacting all past students who participated of this study and then ask them another form to see if after graduation their knowledge improved as dentists or even periodontists. Also new technologies like e-books or video classes might be insert in dentistry college programs, as well as more classes regarding periodontal medicine and the importance of oral-systemic health knowledge among future dentists worldwide.”  

9. Concern of the reviewer

Suggest you check the possibility of adding a few recent references along with others, which may be useful in modifying your paper in the discussion section more appropriately.

Our response: Dear Reviewer #2, we appreciate your suggestion. The references were carefully added. 

Revised text:Page 13, lines 444-451,Chaturvedi, S.; Elmahdi, A.E.; Abdelmonem, A.M.; Haralur, S.B.; Alqahtani, N.M.; Suleman, G.; Sharif, R.A.; et al. Predoctoral dental implant education techniques-students' perception and attitude. J Dent Educ 2021, 85(3):392-400./ Abouzeid, H.L.; Chaturvedi, S.; Abdelaziz, K.M.; Alzahrani, F.A.; AlQarni, A.A.S.; Alqahtani, N.M. Role of Robotics and Artificial Intelligence in Oral Health and Preventive Dentistry - Knowledge, Perception and Attitude of Dentists. Oral Health Prev Dent 2021, 19(1):353-363./ Reddy, L.K.V.; Madithati, P.; Narapureddy, B.R.; Ravula, S.R.; Vaddamanu, S.K.; Alhamoudi, F.H.; Minervini, G.; Chaturvedi, S. Perception about Health Applications (Apps) in Smartphones towards Telemedicine during COVID-19: A Cross-Sectional Study. J Pers Med 2022, 12(11):1920.”  

Round 2

Reviewer 2 Report

Dear author, congratulations for submitting your revised manuscript with all clarifications.